# Impact of Polyethylene Terephthalate Microplastics on *Drosophila melanogaster* Biological Profiles and Heat Shock Protein Levels

**DOI:** 10.3390/biology13050293

**Published:** 2024-04-25

**Authors:** Simran Kauts, Yachana Mishra, Mahendra P. Singh

**Affiliations:** 1Department of Zoology, School of Bioengineering and Biosciences, Lovely Professional University, Jalandhar 14411, India; simran.sh98@gmail.com (S.K.); yachanamishra@gmail.com (Y.M.); 2Department of Zoology, Deen Dayal Upadhyaya Gorakhpur University, Gorakhpur 273009, India; 3Centre of Genomics and Bioinformatics (CGB), Deen Dayal Upadhyaya Gorakhpur University, Gorakhpur 273009, India

**Keywords:** *Drosophila*, toxicity, heat shock proteins, oxidative stress, cell damage, reproductive capacity, microplastic, environmentally relevant

## Abstract

**Simple Summary:**

The threat posed by microplastic toxicity to organisms is growing substantially, and the ramifications of daily microplastic usage cannot be disregarded. The extent of toxicological research pertaining to microplastics has increased due to the grave and alarming nature of microplastic pollution. We have, therefore, conducted research in an effort to determine the toxicological impact of microplastics on the cellular and genetic levels. The toxicity of accumulated polyethylene terephthalate microplastics on *Drosophila melanogaster* has been determined by our research. At higher concentrations of microplastics, cellular and reproductive toxicities have been observed, which correspond to elevated oxidative stress, identified through the analyses of various oxidative stress markers’ activities. Furthermore, the levels of heat shock proteins have been identified, contributing to the understanding of the primary defense mechanism against the toxicity of microplastics. The study has provided significant and concerning insights into the escalating health risks posed by microplastics. It appears that microplastics are inducing genetic alterations; therefore, further investigation should be undertaken at the genetic level to clarify the potential transgenerational consequences that pose a significant risk to future generations.

**Abstract:**

Microplastics and nanoplastics are abundant in the environment. Further research is necessary to examine the consequences of microplastic contamination on living species, given its widespread presence. In our research, we determined the toxic effects of PET microplastics on *Drosophila melanogaster* at the cellular and genetic levels. Our study revealed severe cytotoxicity in the midgut of larvae and the induction of oxidative stress after 24 and 48 h of treatment, as indicated by the total protein, *Cu-Zn SOD*, *CAT*, and MDA contents. For the first time, cell damage in the reproductive parts of the ovaries of female flies, as well as in the accessory glands and testes of male flies, has been observed. Furthermore, a decline in reproductive health was noted, resulting in decreased fertility among the flies. By analyzing stress-related genes such as *hsp*83, *hsp*70, *hsp*60, and *hsp*26, we detected elevated expression of *hsp83* and *hsp70*. Our study identified *hsp83* as a specific biomarker for detecting early redox changes in cells caused by PET microplastics in all the treated groups, helping to elucidate the primary defense mechanism against PET microplastic toxicity. This study offers foundational insights into the emerging environmental threats posed by microplastics, revealing discernible alterations at the genetic level.

## 1. Introduction

Plastics are essential elements in contemporary society and play a crucial role in several aspects of our daily lives. One of the positive attributes associated with plastic, such as its longevity, has posed challenges in the context of plastic waste management. The prolonged utilization of these materials, coupled with the significant proportion of approximately 40% being composed of single-use items, implies a consistent increase in the quantity of waste generated [1]. The impacts of both macro- and microplastics on ecology and physiology are well documented, and comprehending the intricate cellular interactions of microplastics is paramount for a thorough evaluation of their biological consequences. Despite heightened awareness of microplastic pollution, there is a significant knowledge gap regarding the specific ways in which microplastics interact with cellular components, including membranes, organelles, and molecular pathways [2]. Hence, unraveling the cellular mechanisms governing microplastic uptake, intracellular behavior, and biological impacts is imperative for guiding risk assessments, formulating effective mitigation strategies, and safeguarding both the environment and human health [3]. Many studies have been conducted on the behavioral, developmental, and physiological changes caused by microplastic accumulation in insects [4], but to identify genetic alterations, more research is required to determine the upcoming health threats.

Presently, the predominant focus on the hazards associated with microplastics is mostly polystyrene (PS) or polypropylene (PP) [5]. Limited research pertaining to several categories of microplastics has been conducted. In light of this rationale, the material chosen for in-depth examination was polyethylene terephthalate, which was subjected to varying concentrations and durations of exposure for the purpose of further exploration. PET is extensively used within the packaging sector, constituting a substantial 71% of the total plastic consumption in Europe [6]. Furthermore, its remarkable ability to resist friction and its mechanical qualities have resulted in its frequent use in the manufacturing of drinkable water bottles. The findings of the survey indicated that PET accounts for 84% of the total composition of water bottles that are reusable and 31% of beverage bottles [7]. However, the widespread adoption of these products has significantly contributed to the global escalation of plastic pollution [8].

Research has shown that microplastic particles have been detected in human excrement, with the most often identified types being PP and PET [9]. Moreover, a group of researchers reported that a collective sum of 12 different microplastic particles was detected inside the placentas of four different women [10]. Although it is well acknowledged that performing research on animals in vivo is the most efficacious approach for obtaining reliable data that can be used for experimental approaches [11], these investigations have notable limitations that may be ascribed to ethical considerations and difficulties related to manipulation, including increased financial costs and time commitments [12]. Upon careful consideration of the aforementioned difficulties, we opted to use a model organism, namely the *Drosophila melanogaster* wild-type strain (Oregon R^+^), to evaluate the potential risks and the role of stress biomarkers, particularly heat shock proteins (HSPs) that are linked to PET microplastics. HSPs are widely recognized for their participation in numerous cellular processes, including protein synthesis, assembly and folding, translocation, degradation, and conformational maintenance [13]. Additionally, HSPs are indispensable for the activation of client proteins within cells. It has also been discovered that under stressful conditions, they aid in protein refolding and membrane stabilization [14]. The selected upregulation of HSPs occurs in response to physiological, pathological, metabolic, or environmental stressors; this function serves as an inherent cellular defense mechanism [15]. The aim of this research is to investigate studies that have not been properly addressed in previous research. The toxicological effect of microplastics has been studied in various organisms, but the defense system of the body after microplastic accumulation is still unclear. Therefore, we are addressing the effects of microplastics and determining the role of HSPs.

The species *D. melanogaster* is easy to culture. *Drosophila* has many advantages for experimental purposes, such as a short life cycle and requiring little ethical permission [16]. Furthermore, its genomic composition includes genes that exhibit homology to almost 75% of the genes associated with human illnesses [17]. Recently, *Drosophila* has been used for evaluating the potential hazards associated with exposure to polystyrene nano- and microplastics [18]. In our research, the possible dangers associated with PET microplastics were examined by investigating cellular and reproductive toxicity and examining the role of stress biomarkers after exposure in *Drosophila*.

## 2. Methods and Materials

### 2.1. Formation of Polyethylene Terephthalate (PET) Microplastics

Plastic pellets of PET were acquired from Sigma-Aldrich (Jalandhar, Punjab, India) and subsequently pulverized into dust-like powder using a grinder. The resulting PET plastic powder was then separated using a 0.02 mm sieve to obtain PET MPs ranging from 2 to 100 µm in size, as characterized in previous research [19].

### 2.2. Culturing of Model Organisms for Examination of the Health Effects of PET MP

This study was conducted in a controlled environment using *Drosophila melanogaster*, especially the Oregon R^+^ wild-type strain. The flies were fed a standard *Drosophila* diet composed of corn flour, propionic acid, agar, yeast, sodium benzoate (Hi-media, Jalandhar, Punjab, India), and sulfur-free sugar [20,21]. The flies were exposed to controlled environmental conditions, characterized by a 12-h light–dark cycle and a temperature that was consistently maintained at 24 ± 1 °C. The aforementioned criteria were maintained inside a laboratory situated at Lovely Professional University in Phagwara, Punjab, India.

### 2.3. Drosophila Treatment Protocol Using PET MP Exposure

Within the experimental framework, there are 5 different groups. Group I was assigned to the control group and was exposed to culture conditions using standard feed for *Drosophila.* Group II, the vehicle control group, contained ethanol and distilled water mixed with food. Groups III, IV, and V were administered a diet containing a mixture of food and PET MP at concentrations of 10, 20, and 40 g/L, respectively, in accordance with the dosages specified in previous research [19,22,23], with slight modifications. A solution comprising ethanol and distilled water (DW) at a ratio of 1:1 for each concentration was used. The larvae of each group were allowed to eat their specific food for 24 or 48 h. The flies were then exposed to a 15-day intervention to assess other effects of PET MP [22].

### 2.4. PET MP Internalization in Drosophila

The ability to perform precise risk assessments in related research is contingent upon the identification of PET MP accumulation in *Drosophila*. The aim was achieved by the use of a focused strategy. The use of Nile red dye staining has emerged as a cost-effective and user-friendly technique for evaluating the adverse ecological effects associated with a wide range of microplastics [24]. The PET MP was stained according to a previously described methodology [25]. A volume of Nile red (Hi media, DJ Corporation, Jalandhar, Punjab, India) solution (1 mL) in 0.50% dimethyl sulfoxide was added. The pellet was stained and subjected to several washes using 0.10 M phosphate-buffered saline (PBS) at pH 7.40 in ethanol solution to eliminate any residual staining. The presence of stained PET MP in the digestive tract of *Drosophila* larvae may be readily observed without any magnification. PET MP particles were detected by confocal microscopy (CLSM, Olympus, FV1200, Bhatinda, India), wherein observations were made on dissected larvae, which were suspended in a 1% solution of PBS and subsequently affixed on a microscopic slide with a single cavity. To perform confocal visualization of Nile red-stained PET MP and confirm their presence, an excitation wavelength of 514 nm was used, and the emitted light was collected within the range of 546–628 nm. With the help of previous publications, we confirmed the green fluorescence of the PET MP [26].

### 2.5. Determination of Cellular Toxicity in the Gut Region (Trypan Blue Staining)

The evaluation of cell viability was conducted according to the protocol mentioned in the publication [27], with some modifications [20]. This expeditious and straightforward technique enables the differentiation between viable and nonviable cells. The evaluation of cell death involves a comprehensive examination of the gastrointestinal tract, and the principle behind this method relies on the impermeability of the cell membrane to blue dye. Living cells possess an intact cell membrane, preventing the passage of trypan blue into the cytoplasm [28]. The cytotoxicity of the PET MP was assessed using trypan blue dye in the tissues of *Drosophila* subjected to treatment. Following the completion of the treatment, a series of washes were performed on a total of 10–12 larvae using a phosphate-buffered saline solution at a concentration of 0.1 M and a pH of 7.4. Subsequently, the midguts that had been dissected were submerged in a solution of trypan blue dye (Hi media, Jalandhar, Punjab, India) (0.4%) for 15 min. The larvae were examined using a stereomicroscope (Quasmo, Kwality Scientific, 220 V AC, 50 Hz, Jalandhar, India), and photographs were captured to facilitate trypan blue staining and meticulous analysis. Using ImageJ software version 5.0, we calculated the percentage of stained cells.

### 2.6. Preparation of Homogenates

To prepare the samples, we carefully dissected the midguts of third instar larvae from different experimental groups, including the control, vehicle control, and 10, 20, and 40 g/L PET groups. The dissected midguts were then crushed in ice-cold phosphate buffer solution (pH 7.4) containing 0.15 M potassium chloride to create a larval homogenate (10%). Homogenized samples were prepared and subsequently subjected to centrifugation at 4 °C for 10 min at a speed of 12,000× *g*/min. The supernatant was collected by passing it through a 10 mm diameter nylon membrane filter. This collected supernatant (homogenate) was further used for various experiments [29,30].

### 2.7. Total Protein Content, Cu-Zn Superoxide Dismutase (SOD) Activity, Catalase (CAT) Activity, and Lipid Peroxidation

The total protein content of PET MP-treated larvae was determined, and for a standard reference of protein bovine serum albumin (BSA), a previously described method [31], a commonly employed protein assay technique was applied [32]. For the determination of cytosolic *Cu-Zn* SOD activity, the abovementioned protocol [33] was followed, with some modifications [34]. The activity of catalase (CAT) in both the control and treated larvae was assessed by using a protocol [35] by measuring the enzyme’s capacity to catalyze the splitting of hydrogen peroxide (H_2_O_2_) during a 1-min incubation period. The experimental procedure used in the study for quantifying malondialdehyde content as an indicator of lipid peroxidation (LPO) was based on a previous methodology [36].

### 2.8. Dye Exclusion Test of the Ovaries and Testes of Drosophila to Determine Reproductive Toxicity

To assess potential tissue damage in the reproductive organs of adult flies, similar to previous procedures [37], the flies were treated with different PET MP concentrations for 15 days. Five to ten ovaries and testes of *Drosophila* from each group were dissected and subjected to staining with trypan blue dye according to the protocol [27,38] for 15 min. Following the staining process, the samples were thoroughly rinsed with phosphate-buffered saline (PBS) two to three times. The organs of 10–15 flies were examined via a stereomicroscope (Quasmo, Kwality Scientific, 220 V AC, 50 Hz), and images were taken to confirm the results.

### 2.9. Fertility, Fecundity, and Reproductive Performance

The approach used in this study was based on methodology [39], with several adjustments [40]. First instar larvae that had hatched after synchronous egg laying for a duration of 0.5 h were subsequently placed in several feeding media. The groups included a standard food medium (control), a vehicle control, and food mixed with varying concentrations of PET MP at 10, 20, and 40 g/L. The larvae were allowed to nurture their surroundings as they progressed through their personal growth. Virgin female and male flies were observed upon emergence from control and treated food sources. The plants were then separated and paired in vials (1 male + 1 female) containing normal food for mating purposes. Five pairs of flies were selected for each treatment group, and they were individually placed in five vials. Over the course of the following ten days, the flies were moved to new vials on a daily basis.

The total number of eggs deposited within this time frame was recorded. Based on the provided data, the total fecundity, which refers to the overall number of eggs laid down throughout a span of 10 days, was determined. We also recorded the total number of flies that emerged from the eggs laid over the course of the ten-day period. By calculating the average number of flies that emerged per pair during this ten-day interval, we obtained a metric to assess reproductive efficacy. Additionally, the fertility percentage was also determined. The results were calibrated using mortality data.

### 2.10. Qualitative RT-PCR Analysis of the Stress Genes hsp83, hs70, hsp60, and hsp26

The gastrointestinal regions of *Drosophila melanogaster* larvae were extracted from all experimental groups after 48 h of treatment. This was achieved by submerging third instar larvae in Poels’ salt solution (PSS). Subsequently, in accordance with previous methods [20], the extracted tissue was transferred to Eppendorf containers that were filled with TRIzol Reagent by using an RNA extraction kit (Bioserve RNA Extraction Kit, Hyderabad, India) to facilitate total RNA extraction. To determine the concentration and purity of the isolated RNA, the absorbance ratios at 260/280 and 230/260 nm were measured with a NanoDrop spectrophotometer (Denovix, Bioserve, Hyderabad, India).

### 2.11. cDNA Synthesis

The RNA that was acquired underwent reverse transcription by cDNA synthesis using superscript IV VILO master mix (Invitrogen by Thermo Fisher Scientific, Karnataka, India) in accordance with the guidelines provided by the manufacturer. Each reaction mixture consisted of total RNA (10 μL), RT buffer for M-MuLV (4 μL), 10x solution for M-MuLV (2 μL), M-MuLV reverse transcriptase (RNase H) (1 μL), ribonuclease inhibitor (0.5 μL), 10 mM dNTP mix (2 μL), and molecular grade water to make a final volume of 20 μL. The synthesized cDNA was stored at −20 °C until further use.

### 2.12. Polymerase Chain Reaction (PCR)

Next, quantitative PCR (qPCR) was performed utilizing a Quant Studio 5 Real-time PCR system (Thermo Fisher Scientific, India) and the previously designed primers [40] *hsp*83, *hsp*70, *hsp*60, and *hsp*26, which are listed in Table 1. The PCR mixture (total 25 μL) consisted of 2X PCR TaqMixture (12.5 μL), 10 μM each of forward and reverse primers, cDNA (2 μL), and molecular biology grade water. The optimized PCR conditions consisted of an initial cycle of 94 °C for 3 min (denaturation), followed by 35 cycles (*hsp*83, *hsp*70, *hsp*60, and hsp26) of 95 °C for 30 s (denaturation), 55 °C for 30 s (annealing), 72 °C for 1 min (extension), and a final step at 72 °C for 5 min (final extension). The amplicons were separated on a 2% agarose gel containing ethidium bromide at 5 V/cm and visualized with a Vilber gel doc imaging system model (E-BOX CX5. TS, Marne-la-Vallée, France). The intensity of the bands and % of gene expression was quantified by ImageJ software. Relative quantification of gene expression was performed in each experimental group using three independent biological replicates, with the concurrent amplification of β-actin serving as an internal control.

## 3. Statistical Analyses

In our study, significant differences were calculated by using the mean ± SEM (*n* = 3). Statistical analysis was performed using one-way ANOVA, and Tukey’s multiple comparison test was conducted using GraphPad Prism software (version 5.01).

## 4. Results

### 4.1. PET MP Internalization in Drosophila

To investigate this crucial step, confocal microscopy was used. Notably, the digestive tracts of the larvae exhibited distinctive green fluorescence attributed to PET, as illustrated in Figure 1. Subsequent analysis involved scrutinizing confocal microscopy images of the midgut portion of the dissected larvae shown in Figure 1A2. The identification of microplastic particles within the midgut of *Drosophila* larvae provides compelling grounds for assessing the potential risks associated with PET microplastics in organisms.

### 4.2. Trypan Blue Assay

The experimental findings are shown in Figure 2. Compared with those in the control group, the midgut tissues of larvae subjected to PET MP at concentrations of 20 and 40 g/L had blue staining at rates of 45% and 61%, respectively. No significant blue staining was observed in the untreated groups.

### 4.3. Total Protein Concentration, Cu-Zn SOD Activity, CAT Activity, and MDA Content

A significant effect of PET microplastics on the enzymatic activity of *Drosophila* was observed. Following exposure to PET MP at concentrations of 20 and 40 g/L, the total protein content in *Drosophila* third instar larvae significantly decreased, as shown in Figure 3A. After 24 h of exposure, the 40 g/L PET group displayed a lower protein content. Similarly, after 48 h, both the 20 g/L and 40 g/L groups showed decreased protein concentrations in comparison to those in the control/vehicular control group. However, no significant decrease in protein concentration was observed in the 10 g/L group after exposure. Figure 3B depicts the observed *Cu-Zn* SOD activity in larvae in the test groups. Compared with those of the control larvae, the SOD activity of the larvae subjected to 20 and 40 g/L PET increased after 24 and 48 h. There was no significant change in SOD activity within the group treated with 10 g/L PET/MP. The data shown in Figure 3C illustrate the catalase activity observed in larvae in the various treatment groups. Compared with the control larvae, the larvae treated with PET MP at concentrations of 20 and 40 g/L exhibited an increase in catalase activity after 24 and 48 h. Catalase activity did not substantially change in the 10 g/L group. Compared with those in the control group, the MDA content in both the 20 and 40 g/L PET groups substantially increased after 24 and 48 h of treatment. No statistically significant alteration in the MDA content was detected in the group treated with 10 g/L PET MP, as shown in Figure 3D.

### 4.4. Cytotoxicity in Reproductive Organs

The cytotoxicity of the PET MP was evaluated by a dye exclusion test (trypan blue) on testis and ovary tissues from treated *Drosophila* flies to determine whether PET MP exposure results in cytotoxic effects. Compared with those in the control group, the ovaries (mature follicles), male testes, and accessory glands of the flies exposed to PET MP at concentrations of 20 and 40 g/L after 15 days of treatment exhibited blue staining in the female ovaries, male testes, and accessory glands, as shown in Figure 4. There was no observed blue stain in the control or vehicle control groups. We also observed a reduced size of the left ovary in females, as shown in Figure 4B5, during our analysis, but this finding is based only on visible observation and needs more scientific anatomical evidence.

### 4.5. Fecundity, Fertility, and Reproductive Performance

A significant decrease in the fecundity of *Drosophila melanogaster* in the 20 and 40 g/L groups was found, compared to the control or vehicle control group. The emergence of flies from egg laying was also decreased in the 20 and 40 g/L PET groups, as depicted in Figure 5A, compared to that in the control group. The fertility percentage shown in Figure 5B indicates that the overall reproductive health of *Drosophila* flies deteriorates after the consumption of high dosages of PET MP. There was no significant effect of PET MP in the 10 g/L PET MP group.

### 4.6. RT-PCR Analysis

Figure 6 shows the gel agarose band images and the fold changes of gene expression in *Drosophila* larvae after 48 h exposure to PET microplastic. The percentages of genes expressed for *hsp*83, *hsp*70, *hsp*60, and *hsp26* in the control, vehicular control, 10, 20, and 40 g/L PET groups were (8.7%, 10%, 18.41%, 20.5%, and 42.5%), (13.4%, 13.8%, 16.3%, 20.43%, and 35.7%), (39.0%, 25.4%, 13.19%, 12.53%, and 9.9%), and (20%, 19%, 21%, 20%, and 20%), respectively. Notably, *hsp*83 exhibited more pronounced overexpression, and these findings collectively identify *hsp*83 as a prominent and responsive biomarker by emphasizing its role as a key component in the initial defense mechanism against microplastic-induced stress. Moreover, we observed that *hsp*60 was downregulated, while *hsp*26 activities did not significantly change compared to the control group.

## 5. Discussion

To obtain comprehensive information on the possible health risks linked to exposure to environmental microplastics or nanoplastics, it is necessary to conduct a thorough investigation. Ingestion is a well-recognized pathway for exposure to micro- and nanoplastic particles. To analyze the detrimental effects of microplastics, confirmation of the accumulation of PET MPs is necessary. Therefore, understanding the interactions between microplastics and the components of the digestive system is important for toxicological research. According to an available study [41], snails subjected to extended exposure to PET microfibers exhibited a reduction in food consumption and excretion, resulting in the impairment of villi in the stomach and intestine. Moreover, it has been shown that this particular kind of intestinal injury has the potential to diminish nutritional absorption in *Daphnia* [42]. In *Drosophila*, gastrointestinal injury was also observed after polystyrene microplastic exposure [43]. Oxidative stress is one of the outcomes linked to exposure to microplastics [44]. These findings are in agreement with previous research [45] showing that after the accumulation of polystyrene microplastics, oxidative stress increases within the liver of *Eriocheir sinensis;* moreover, antioxidant enzyme activities in *Brachionus calyciflorus* change after being exposed to polystyrene microplastics, and increased SOD and CAT activity is observed [46].

The reproductive health of the flies was found to be impaired, which is a novel topic for future research. In female flies, cellular toxicity in the mature follicles of the ovaries can be responsible for changes in or effects on reproductive health because there are several variables that may be connected to this toxicity, such as female fertility, fecundity, the egg distribution chamber, embryo development, yolk protein-encoding gene expression, meiotic crossing over, and maternal protein localization [47]. In male flies, toxicity occurring in the accessory gland and testis could also lead to changes in sperm morphology, sperm formation, sperm count, courtship behavior, and sperm motility [48]. For the first time, a recent study revealed the presence of PVC, PS, and PE microplastics in human testis and semen samples [49]. The accumulation of polystyrene microplastics in *Danio rerio* (zebrafish) decreased the fertility rate and had a severe effect on gonad morphology [50]. Oral ingestion of microplastics in mice leads to infertility, reduction in fertilization, and embryo development; furthermore, during the gestation and lactation periods, exposure to microplastics results in transgenerational microplastic reproductive toxicity [51]. The hazardous effects that were determined by our research on reproductive health could cause alterations in the expression of genes that are responsible for reproduction, so further research at the molecular level using transcriptomic analysis or any scientific method is needed to determine the variation in genes. When organisms consume microplastics, they can accumulate and be passed on to their offspring, leading to transgenerational or multigeneration toxicity [52], as shown in the schematic representation in Figure 7.

Life cycle assessments indicate that reproductive health alteration and toxicity may manifest during various stages, such as gamete production, embryogenesis, hatching, secondary maturation, or the transformative processes involved in these crucial biological activities [53]. A study determined that the presence of polyethylene microplastics in the soil at a concentration of 0.5% resulted in a 70% decrease in earthworm reproduction for both the parent (F0) and first filial (F1) generations compared to soil without microplastic contamination. Additionally, notable DNA damage was identified in the F0 generations after a 28-day period [54].

At the molecular level, the expression of the *hsp*83 and *hsp*70 genes was upregulated. Likewise, the upregulation of hsp70 expression was detected in the giant river prawn *Macrobrachium rosenbergii* subjected to microplastics composed of polystyrene and polyethylene [55]. The heat shock response facilitated by heat shock transcription factor 1 (HSF1) was impeded by a mixture of di-(2-ethylhexyl) phthalic acid (DEHP) and polypropylene microplastics [56]; this combination induced neurotoxicity in immature mice via neuronal apoptosis and neuroinflammation. While many studies have examined the effects of different microplastics and stress genes, the precise *hsp* that serves as an early indicator of cellular redox changes and can be used to detect microplastic toxicity remains uncertain. However, our research has now produced conclusive results at the gene level regarding this aspect. Elevated levels of heat shock proteins (HSPs) are a defensive reaction to stress [57], and moreover, *hsp*26, *hsp*90, and *hsp*70 are stress-inducible HSPs that have been the subject of extensive research [58]. Previous studies have demonstrated that malignancy is characterized by a substantial increase in the expression and activity of these chaperones, which are also susceptible to various stimuli that induce cell death [59].

Under normal conditions, HSF-1, a transcription factor, is retained in the cytoplasm by HSPs such as *hsp*90 and *hsp*70, which bind to HSF1, preventing its transcriptional activity [60]. However, according to our study, we believe that the mechanism for stress gene regulation in response to stress induced by PET microplastics is as follows: HSPs detach from this complex, activating HSF1. Once activated, HSF1 moves into the nucleus and binds to specific sequences called heat shock elements (HSEs) located upstream of heat shock gene promoters, thus initiating the transcription of HSP genes, as illustrated in Figure 8. Our study sheds light on the toxicity of microplastics through the involvement of different *hsp* genes in defense mechanisms against PET microplastics. Ultimately, our findings suggest that the overexpression of specific stress genes could indicate impending health threats from microplastics, serving as early indicators of toxicity [59]. Numerous studies have highlighted metabolic disturbances, neurotoxicity, and increased cancer risk in humans following exposure to microplastics [61]. Whether or not the same stress gene functions as a stress biomarker for all forms of microplastics remains unknown; therefore, additional comparative research between various microplastics and stress genes is necessary to answer this question.

Therefore, the present research should further investigate genotoxicity because when assessing the potential adverse consequences of environmental contaminants, genotoxicity has emerged as an essential biomarker. DNA damage is well recognized for its significant impact on several health outcomes, including but not limited to gene/chromosome mutations, carcinogenesis, and aging [62]. Genotoxicity is often regarded as a surrogate biomarker for the process of carcinogenesis, and it plays a significant role in both the initial and advancement stages [63]. Despite the significance of biomarkers, few studies have assessed the possible genotoxic impacts of PET MP. The results of our study provide initial data that might serve as a foundation for further investigations into the effects of PET MP on *Drosophila*. Additional investigations are required to explore the transgenerational impacts and ascertain the extent to which an organism’s body exhibits resistance to microplastic toxicity. If such resistance exists, it is crucial to understand the specific level at which it manifests and to elucidate the physiological or genetic changes that occur within the organism’s body to facilitate this.

## 6. Conclusions

In general, our research indicated that the toxicity of PET MPs is dose-dependent. Specifically, we observed that an increase in plastic concentration corresponds to a heightened toxicological impact. At elevated concentrations, PET MPs induced significant cytotoxicity, oxidative stress, and reproductive damage in the model organism *Drosophila melanogaster*. Furthermore, our study also determined the expression of stress gene biomarkers responsible for microplastic-induced toxicity. With regard to the population, our research results suggest that a certain plastic concentration is permissible for use. Nevertheless, due to the pervasive nature of plastic in our everyday existence, it is imperative that we not completely dismiss its significance. Nevertheless, when the concentration is beyond a certain level, it becomes a matter of significant apprehension. The analysis of the underlying toxicity suggested that plastic has a persistent and incremental impact on daily life. Furthermore, investigating how microplastics impact the physiology of insects is crucial. This is because insects are considered potential natural plastic degraders. Recent research using advanced genetic techniques has suggested that *Drosophila melanogaster* could serve as a model organism for developing super degrading insects. The findings of this study provide valuable insights into the significant consequences of microplastic contamination and emphasize the urgent need for the implementation of efficient mitigation measures. Hence, more investigations are necessary to assess the impact of microplastics at the molecular level.

## Figures and Tables

**Figure 1 biology-13-00293-f001:**
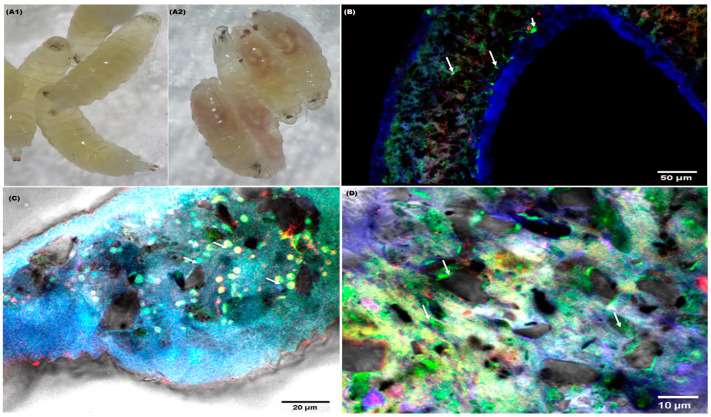
Detection of PET MP by using confocal microscopy. (**A1**) Larvae of the control group, (**A2**) larval gut treated with Nile red-stained PET MP, (**B**–**D**) confocal images of (**A2**) larval midgut showing the accumulation of PET MP with green fluorescence, as represented by white arrows. For (**A1**,**A2**), a stereomicroscope was used to capture images.

**Figure 2 biology-13-00293-f002:**
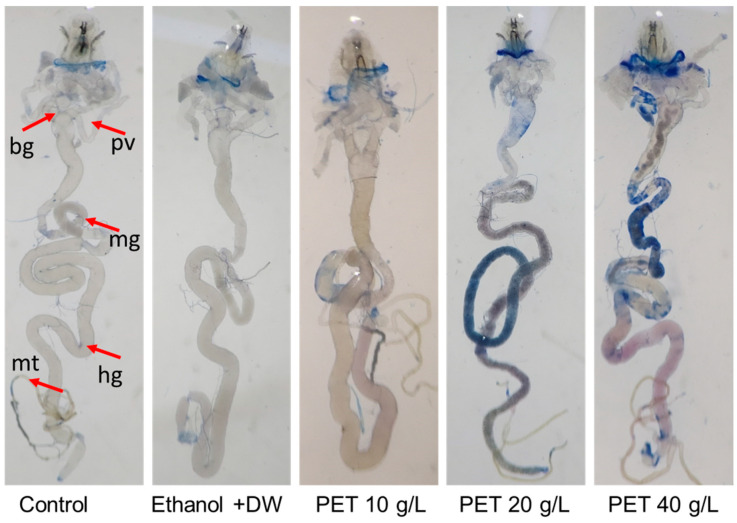
The dye exclusion test was conducted using trypan blue staining on dissected 72 h third instar larvae of *Drosophila melanogaster* (Oregon R^+^) in the control or vehicle control group and larvae exposed to 10 g/L, 20 g/L, or 40 g/L PET. The staining percentage was 1% in the control group, 2% in the ethanol + DW group (the vehicle control group), 12% in the 10 g/L group, 45% in the 20 g/L group, and 61% in the 40 g/L group. bg = brain ganglia, pv = proventriculus, mg = midgut, hg = hindgut, mt = Malpighian tubules.

**Figure 3 biology-13-00293-f003:**
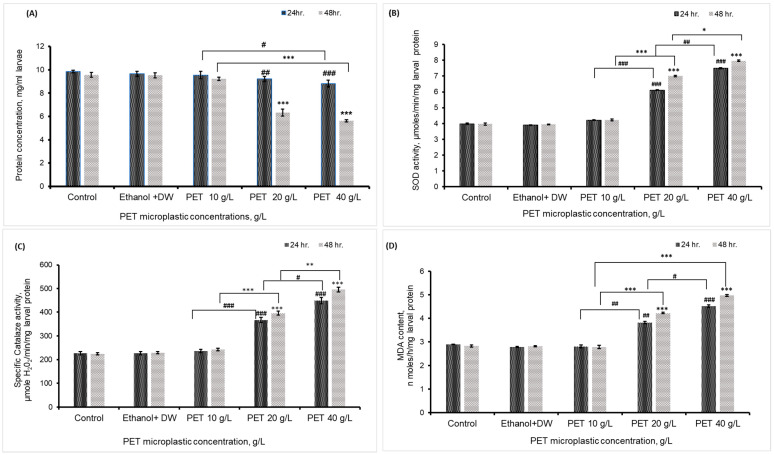
(**A**) Total protein content, (**B**) Cu-Zn SOD activity, (**C**) catalase activity, and (**D**) MDA content of third instar *Drosophila* larvae after 24 h and 48 h of treatment with PET MP. The average ± SEM (*n* = 3) was calculated, and the significant differences were considered at ^##^ *p* < 0.01 and ^###^ *p* < 0.001 after 24 h of treatment. Similarly, *** *p* < 0.001 for the 48 h treatment group compared with the control group. ^#^*p* < 0.05 illustrate significance difference for 24 h treatment between PET microplastic treated groups, similarly * *p* < 0.05, and ** *p* < 0.01 indicate significant difference for 48 h treatment between the PET microplastic treated groups.

**Figure 4 biology-13-00293-f004:**
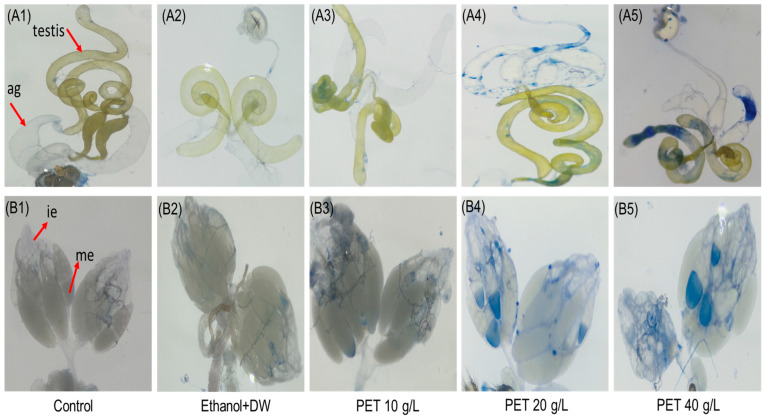
The cytotoxicity test involving trypan blue staining of *Drosophila* flies subjected to 15 days of treatment. The groups included controls, vehicle controls, and those with PET concentrations of 10, 20, and 40 g/L. (**A1**–**A5**) depict male reproductive organs across all test groups, while (**B1**–**B5**) show female reproductive organs. ag = accessory glands, ie = immature eggs, me = mature eggs.

**Figure 5 biology-13-00293-f005:**
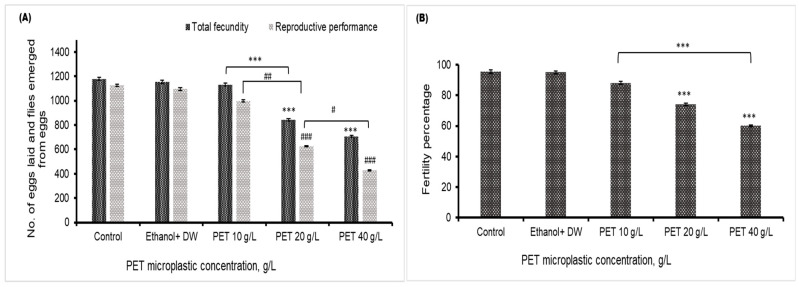
The reproductive health of *Drosophila melanogaster* is depicted in (**A**), which illustrates fecundity (number of eggs) and reproductive performance (total number of emerged flies), presented as the mean values with standard error of the mean (SEM) (*n* = 3). Statistical significance is denoted by *** *p* < 0.001 for fecundity compared to the control group and ^###^ *p* < 0.001 for reproductive performance relative to the control group. ^#^ *p* < 0.05 and ^##^ *p* < 0.01 indicate significance difference between PET microplastic treated groups (**B**) represents the fertility percentage, with statistical significance indicated by *** for *p* < 0.001 compared to the control.

**Figure 6 biology-13-00293-f006:**
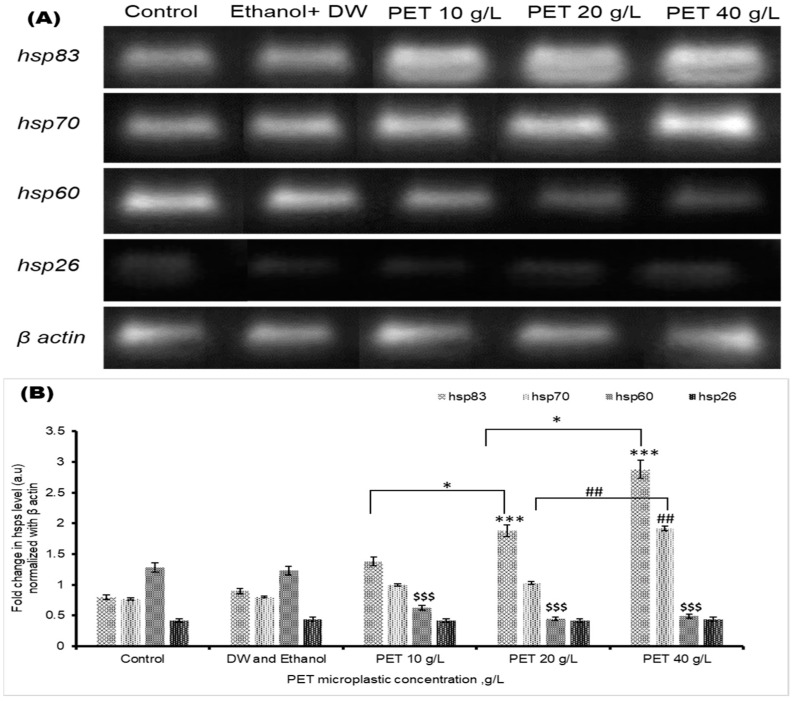
(**A**) Quantitative analysis via RT-PCR for the specific genes *hsp*83, *hsp*70, *hsp*60, and *hsp*26 on agarose gels. (**B**) The graph shows the fold change in the gene expression level normalized to that of β actin, and statistical significance was assigned as *** *p* < 0.001 for *hsp*83, ^##^ *p* < 0.01 for *hsp*70, and ^$$$^ *p* < 0.001 for *hsp*60 in comparison to the control group and * *p* < 0.05 for *hsp*83 in comparison between the PET microplastic treated groups.

**Figure 7 biology-13-00293-f007:**
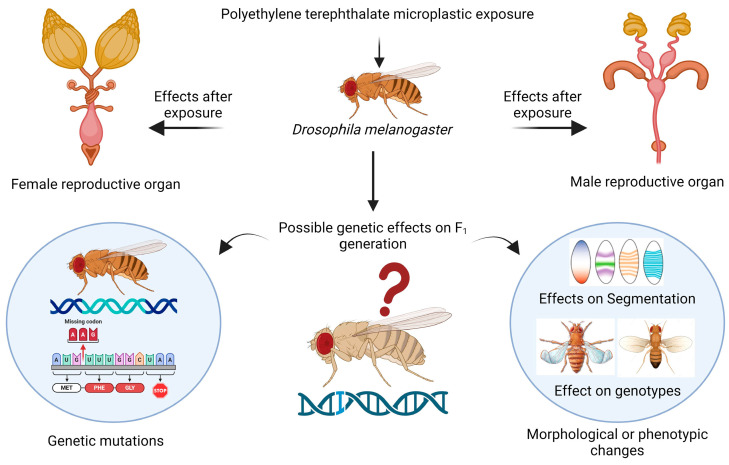
Schematic representation of possible transgenerational effects on *Drosophila melanogaster* after exposure to PET microplastics. Created with BioRender.com.

**Figure 8 biology-13-00293-f008:**
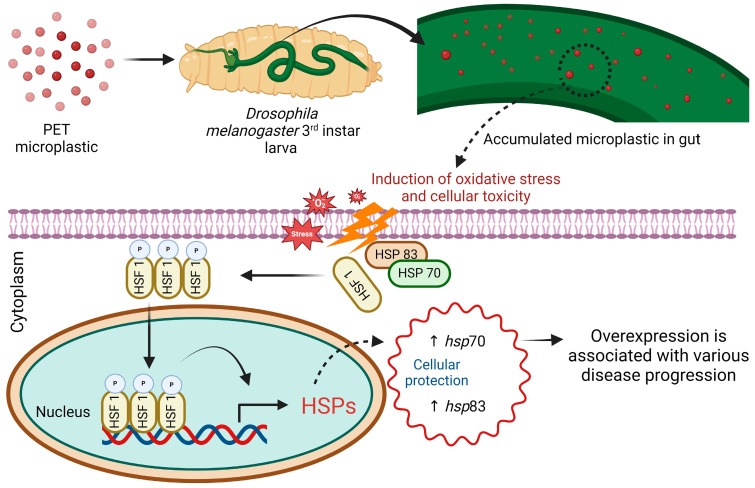
Schematic representation of the accumulation of PET microplastics and the role of heat shock proteins in promoting cellular protection for cell survival.

**Table 1 biology-13-00293-t001:** Both forward and reverse HSP primers with their sequencing.

*hsp*83 Forward	5′CCCGTGGCTTCGAGGTGGTCT3′
*hsp*83 Reverse	5′TCTGGGCATCGTCGGTAGTCATAGG3′
*hsp*70 Forward	5′GAACGGGCCAAGCGCACACTCTC3′
*hsp*70 Reverse	5′TCCTGGATCTTGCCGCTCTGGTCTC3′
*hsp*60 Forward	5′CCTCCGGCGGCATTGTCTTC3′
*hsp*60 Reverse	5′AGCGCATCGTAGCCGTAGTCACC3′
*hsp*26 Forward	5′CAAGCAGCTGAACAAGCTAACAATCTG3′
*hsp*26 Reverse	5′GCATGATGTGACCATGGTCGTCCTGG3′
β actin Forward	5′CCTCCGGCGGCATTGTCTTC3′
β actin Reverse	5′GGGCGGTGATCTCCTTCTGC3′

## Data Availability

Data contained within the manuscript.

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
