# Peer review of "Impact of Polyethylene Terephthalate Microplastics on Drosophila melanogaster Biological Profiles and Heat Shock Protein Levels"

_biology, 2024, doi:10.3390/biology13050293_

Round 1
Reviewer 1 Report
Comments and Suggestions for Authors
This is a nice work on the effect of MPs on the physiology of an animal model organism. While the topic is very fashionable and relevant, I found that the methodology is not accurate and controls are lacking in some experiments.
l72 "....detected inside the 4 different women’s placenta of [10]"
l80 capitalize hsps)
l88-90 these two sentences are a bit redundant and express the concept in a twisted way
l92 "condensed life cycle," short life cycle?
l92 what do you mean "easily manageable alteration"?
l93 "acquisition of ethical permission" no ethical permission is required to perform experiments on insects. reference 16 does not apply to this topic.
l121 typo (samll)
l230-231 "...we have used an experimental method-230 ology called confocal microscopy, simply say we have used confocal microscopy
l234"...the emitted light was collected within the range of 546-628 nm" Why a so wide wavelength range?
confocal microscopy methods details should be given in the methods section
Figure 1. Larvae in panel A1 appear morphometrically different from larve showed in panel A2. Are they different strains?
Figure 1. Panels B-D are not convincing. How were these images produced? The authors should give more details. What part of the gut are they focusing on? What are different colors? Also I need to see a control here.
It is not clear at all how MPs become fluorescent.
l249 ImageJ methods should be described and reported in the appropriate section of the manuscript
Bar chart are not sufficient to support the results. Raw data must be associated to the manuscript as supplementary files. This is mandatory in order to make the manuscript acceptable for publication
Primers used for RT-PCR analyses should be double checked. Hsp70 primer seems to be non specific (multiple fragments can be amplified), whereas hsp26 and b-actin primers do not anneal at all on the respective targets.
The description of the RTPCR experiments is not accurate. Please give as much as details you can (i.e. cDNA input, amplification reaction conditions, quantification method, raw values....)
Which drosophila actin gene has been used as control?
Figure 6. It seems that the agarose gel picture has been obtained by copy/pasting individual bands. Constructed panels like this are not acceptable. Please show original gel pictures instead (they can be added as supplementary files)
In the discussion (or in the conclusion section) it is worth mentioning that the effect of microplastics on the physiology of insects is an important topico since insects have been proposed as natural plastic degraders (see doi:10.1016/j.chemosphere.2022.133600). Also D. melanogaster has been recently proposed as a model organismi to develop super-degraders by means of modern genetics techniques (see doi:/10.1016/j.scitotenv.2024.169942). The deleterious effect of microplastics ingestion may affect the performances of both natural and transgenic degraders.
Comments on the Quality of English LanguageThe text needs editing
Author Response
This is a nice work on the effect of MPs on the physiology of an animal model organism. While the topic is very fashionable and relevant, I found that the methodology is not accurate and controls are lacking in some experiments.
Response to reviewer: Thank you so much for your valuable suggestions, your opinions are making this manuscript more understandable and improving its scientific values. Moreover, we are very pleased to resolve your all queries.
l72 "....detected inside the 4 different women’s placenta of [10]"
Response to reviewer: Thank you so much for your suggestion, we apologise for such mistakes. We have changed this phrase of sentence according to the suggestion. please verify the correction made in line 78 and 79
l80 capitalize hsps)
Response to reviewer: Thank you for your valuable suggestion we have capitalize Hsps as per your suggestion, please verify correction in line 86
l88-90 these two sentences are a bit redundant and express the concept in a twisted way
Response to reviewer: Thank you for your keen observation, we kindly want you to verify the statement that has been rephrased in line 94- 99
l92 "condensed life cycle," short life cycle?
Response to reviewer: Yes, the meaning of the statement is short life cycle, we have changed this phrase in the manuscript for better understanding. we request to the reviewer to kindly verify correction in line 101
l92 what do you mean "easily manageable alteration"?
Response to reviewer: we thank reviewer to make us attentive for this mistake, this phrase should not be here in the manuscript. so, we have removed this phrase , kindly verify the correction in line 101.
l93 "acquisition of ethical permission" no ethical permission is required to perform experiments on insects. reference 16 does not apply to this topic.
Response to reviewer: We are grateful to the reviewer for identifying sentences that are pertinent to the references. We would appreciate the reviewer's confirmation that the reference supplied is used to cite the entire statement. The authors of this article indicated the following. To make this statement more accurate, however, we have made a few minor adjustments. Kindly validate the correction made to line 101.
l121 typo (samll)
Response to reviewer: We apologise for making minor typing errors. Correction has been made in the whole manuscript. We request you to verify correction in line 133
l230-231 "...we have used an experimental method-230 ology called confocal microscopy, simply say we have used confocal microscopy
Response to reviewer: We totally agree with this suggestion, thank you so much for this valuable suggestion. We request you to verify the correction in line 272
l234"...the emitted light was collected within the range of 546-628 nm" Why a so wide wavelength range?
Response to reviewer: We appreciate the reviewer for asking this scientific aspect of the experimentation. We kindly want to inform the reviewer that mentioned range is used with the help of previous publication for the protocol in our experiment (Alaraby, M.; Villacorta, A.; Abass, D.; Hernández, A.; Marcos, R. The hazardous impact of true-to-life PET nanoplastics in Drosophila. Sci. Total Environ, 2023, 863, 160954. https://doi.org/10.1016/j.scitotenv.2022.160954.)
confocal microscopy methods details should be given in the methods section
Response to reviewer: Thank you so much for this valuable suggestion. We request you to kindly verify the Methodology of confocal microscopy which is provided in methodology section under (PET MP internalization in Drosophila) in line 141-157
Figure 1. Larvae in panel A1 appear morphometrically different from larvae showed in panel A2. Are they different strains?
Response to reviewer: We really like to appreciate your concern about the morphological characteristics of larvae. We are pleased to resolve your concern that larvae are of same strain, this may be due to the shrinkage of body after a longtime exposure under microscope to capture pictures. we are providing you another image for your concern, please take a look on the below image.
Figure 1. Panels B-D are not convincing. How were these images produced? The authors should give more details. What part of the gut are they focusing on? What are different colors? Also, I need to see a control here.
Response to reviewer: Thank you so much for asking your queries ,we will be very pleased to resolve all of them. We would like to inform you that Panel B-D are the image of larvae midgut captured with different magnifications. As per your suggestion we have incorporated the part of gut in the manuscript please varify changes in line 275 and 276.
For the control group, Figure (A1) is control which do not contain any dye or dyed plastic particle, so they could not undergo for confocal microscopy because do not have any fluorescing agent/dye in it, that is why we have used whole larvae for control group, those were fed with their standard diet only. But, we could use dissected gut image for control if you prefer. We have demonstrated schematic representation of confocal microscopy in below figure for your better understanding about how image produced and pressence of different colours are there. We request you to verify all the necessary references and full methodology is provided in the manuscript in M&M section from line 140- 157.
As this experiment is outsourced, overlaped image is used as result of confocal microscopy which is refelecting the final (dominating) colour of fluorescent particle after overlaping. To support the colour of the PET fluroscent particles we have also cited the previous publication for reference also in line 157.
It is not clear at all how MPs become fluorescent.
Response to reviewer: We really appreciate the reviewer’s query which is very essential for the understanding of experimentation. We will be very pleased to explain the reason of fluorescent of MP.
By using Nile red dye which is fluorescing and hydrophobic in nature, microplastic become fluorescent. Nile red has recently emerged as a rapid, more accessible, and less subjective technique for microplastic quantification. Nile red has quantified microplastics in samples from aquatic, sedimentary, and biological samples. It has also quantified microplastics in bottled water, the findings of which are highly relevant to human health (DOI: 10.1021/acs.estlett.9b00499). So first we have dyed the plastic particles and then washed them thoroughly to remove extra stain as mentioned in methodology. Then stained particles were used for the treatment of microplastic accumulation.
l249 ImageJ methods should be described and reported in the appropriate section of the manuscript
Response to reviewer: Thank you so much for this suggestion, it will make the manuscript well-constructed. Suggestion taken into consideration, please verify the changes in line 173
Primers used for RT-PCR analyses should be double checked. Hsp70 primer seems to be non specific (multiple fragments can be amplified), whereas hsp26 and b-actin primers do not anneal at all on the respective targets.
Response to reviewer: We are very thankful to the reviewer for being a part of this process in making this manuscript more accurate and meaningful. Very keen observation and concerns about every concept is helping in improving the quality of manuscript. We kindly like to inform you that we have double checked the primers according to your suggestion, in previous publication, we have used same sequencing of primers as you can verify from the mentioned reference (Singh, M. P., Reddy, M. K., Mathur, N., Saxena, D. K., & Chowdhuri, D. K. (2009). Induction of hsp70, hsp60, hsp83 and hsp26 and oxidative stress markers in benzene, toluene and xylene exposed Drosophila melanogaster: role of ROS generation. Toxicology and applied pharmacology, 235(2), 226-243.
The description of the RTPCR experiments is not accurate. Please give as much as details you can (i.e. cDNA input, amplification reaction conditions, quantification method, raw values....)
Response to reviewer: Thank you for identifying this important point in the manuscript, we totally agree with your suggestion. So, we have incorporated more information and experimentation details about the RTPCR as per your suggestion. Please verify the changes in line number 239- 260
Which drosophila actin gene has been used as control?
Response to reviewer: Thank you for asking such experimentation queries. To resolve your query we like to inform you that β actin housekeeping gene was used as control and it is also mentioned in table 1
Figure 6. It seems that the agarose gel picture has been obtained by copy/pasting individual bands. Constructe panels like this are not acceptable. Please show original gel pictures instead (they can be added as supplementary files) Bar charts is not sufficient to support the results. Raw data must be associated to the manuscript as supplementary files. This is mandatory in order to make the manuscript acceptable for publication.
Response to reviewer: We would like to extend our sincere gratitude for your valuable suggestions, each of which we find to be extremely precise and beneficial. We are pleased to inform you that the section editor has received all the supplementary data via email, as requested prior to the review process. If further information be needed, we will certainly include it.
In the discussion (or in the conclusion section) it is worth mentioning that the effect of microplastics on the physiology of insects is an important topico since insects have been proposed as natural plastic degraders (see doi:10.1016/j.chemosphere.2022.133600). Also D. melanogaster has been recently proposed as a model organismi to develop super-degraders by means of modern genetics techniques (see doi:/10.1016/j.scitotenv.2024.169942). The deleterious effect of microplastics ingestion may affect the performances of both natural and transgenic degraders.
Response to reviewer: Thankyou so much for this valuable suggestion as this suggestion seems very interesting for the concept of further research. so, we have added this information in the conclusion section in line 500- 503. Please verify the changes.

Reviewer 2 Report
Comments and Suggestions for Authors
The authors study a globally important problem: the impact of microplastics on insects. Unfortunately, there are not enough good publications on this topic. The peer-reviewed article includes research at the level of anatomy, histology, biochemistry and genetics. This makes the manuscript especially interesting. Apparently, the experiment was carried out correctly, and the authors describe reliable scientific results. The disadvantage of the manuscript is the inattentive attitude of the authors to their own results. In diagrams and photographs, many details are indistinguishable.
Disadvantages of the manuscript:
1. In the title of the article it is better to remove “the biological activities, potential transgenerational risks posed to” and “a comparative analysis of levels of”.
2. The abstract is uninformative. It is better to reduce its introductory and methodological parts by 30%. The results in the abstract need to be expanded by 80%.
3. It is better to remove from the keywords the terms that are present in the title of the article. It is necessary to expand the composition of keywords to 8 phrases.
4. Sources in brackets are placed carelessly, for example line 121.
5. Each drug (for example, a dye) after the first mention in parentheses must have an indication of the company, city, country (lines 103-225). The same applies to computer programs and laboratory equipment.
6. It is better to delete the formula on line 204. A description of the calculation process in text form will suffice.
7. Line 223-225: The authors did not indicate that they were calculating the standard error of the mean.
8. The results cannot contain elements of introduction (for example, lines 228-229) and methodology (for example, lines 230-231, 235-237, 243-246, 249). This applies to the entire Results section. Results cannot contain references to literature (line 235).
9. The image in Figures B, C, D needs to be enlarged. You should also label the most important elements in your photos so that readers understand the context.
10. Remove artificial effects on the edges of photos (Figure 2). It's not scientific. Do not use bold fonts for captions of Figures 2-8.
11. The authors studied many dozens of preparations of the Drosophila digestive system. The authors saw general patterns of changes in the length and diameter of different parts of the intestine, accessory glands, and excretory system. The reader will not be able to see all this only based on the analysis of Figure 4. I recommend that the authors, in the title of Figure 4, briefly describe how exactly this version of the experiment differs from the control, indicating with numbers the corresponding changes in the photograph. This drawing is worthy of increasing the image size by 100% in both length and width.
12. In the name of both the abscissa and ordinate axis (Fig. 3), you need to put a comma before the unit of measurement, rather than write (g/l) in parentheses. The same remark applies to the labels on the x-axis of Figure 3A. Don't use bold fonts. There is no need to repeat the legend 4 times.
13. The same remark on Figure 5 and subsequent ones.
14. All % in the article (for example, lines 319-321; in addition, three technical errors were also found on these drains) must be rounded to the nearest tenth.
15. Genes are written in italics (for example, lines 313, 318, 322 and others).
16. Statistical processing of samples must be carried out again. For example, in Figure 6B. The reader wants to see all the differences between all samples at the 0.05 level. A Tukey test needs to be done here. The authors mention it on line 225, but the results are not reflected in Figures 3, 5, 6.
17. Section The discussion should be divided into subsections. This will make the presentation clearer.
18. Figure 8 contains unconfirmed patterns. Be careful when reviewing the literature and displaying the results in a diagram.
19. There is definitely no need to list the diseases of mammals in the picture depicting a fly larva.
20. In literature, semicolons should be between authors.
21. The periods after the abbreviated names of the magazines are placed carelessly. Not all journal names are abbreviated according to the rules.
22. Spaces between initials are placed carelessly.
23. Many articles do not have DOIs.
24. Journal titles and volumes are not in italics.
Author Response
Reviewer #2
The authors study a globally important problem: the impact of microplastics on insects. Unfortunately, there are not enough good publications on this topic. The peer-reviewed article includes research at the level of anatomy, histology, biochemistry and genetics. This makes the manuscript especially interesting. Apparently, the experiment was carried out correctly, and the authors describe reliable scientific results. The disadvantage of the manuscript is the inattentive attitude of the authors to their own results. In diagrams and photographs, many details are indistinguishable.
Response to reviewer: We are very thankful to the reviewer that has provided quite helpful suggestions, and the ideas are very important in terms of making the manuscript more engaging and improving its overall quality.
Disadvantages of the manuscript:
- In the title of the article it is better to remove “the biological activities, potential transgenerational risks posed to” and “a comparative analysis of levels of”
Response to reviewer: Your esteemed recommendation is greatly appreciated. The suggestion provided is more pertinent than the previous one. In accordance with your recommendation, the title of the manuscript has been revised and new title is “Adverse effects of PET microplastics on biological profiles of wildtype Drosophila melanogaster (Oregon R+) and expression of heat shock proteins (hsps)”.
.
- The abstract is uninformative. It is better to reduce its introductory and methodological parts by 30%. The results in the abstract need to be expanded by 80%.
Response to reviewer: Your esteemed suggestion is greatly appreciated. We completely concur with your recommendation that it should include results that are thoroughly explained. Revisions have been implemented to the abstract. We request that you validate the modifications made to lines 18–26 and 29–31.
- It is better to remove from the keywords the terms that are present in the title of the article. It is necessary to expand the composition of keywords to 8 phrases.
Response to reviewer: Thank you for your valuable suggestion. We have incorporated your suggestion. Please verify the changes in line 32, 34
- Sources in brackets are placed carelessly, for example line 121.
Response to reviewer: We apologise for this act of carelessness. We kindly want to inform you that we have used sources repeatedly as per the requirement but corrected the sources of line 134 as they were not in order, please verify the correction.
- Each drug (for example, a dye) after the first mention in parentheses must have an indication of the company, city, country (lines 103-225). The same applies to computer programs and laboratory equipment.
Response to reviewer: We really appreciate reviewer 2 for noticing such points which are very important for a research article. We have incorporated the chemical and instruments information wherever it was not mentioned before in whole manuscript. Please verify the changes in line 112, 120, 146, 151, 170, 205, 234, 236,
- It is better to delete the formula on line 204. A description of the calculation process in text form will suffice.
Response to reviewer: Thank you for the suggestion, we have removed the formula as per your suggestion, please verify in line 226.
- Line 223-225: The authors did not indicate that they were calculating the standard error of the mean.
Response to reviewer: Thank you for this excellent suggestion, statistical analysis needs every information about the experimentation so, we have added more information in line 273- 275. Please verify the changes.
- The results cannot contain elements of introduction (for example, lines 228-229) and methodology (for example, lines 230-231, 235-237, 243-246, 249). This applies to the entire Results section. Results cannot contain references to literature (line 235).
Response to reviewer: We deeply appreciate the insightful suggestions provided by reviewer 2 regarding the manuscript's strength and significance. We extend our sincere apologies for any inconsistencies that may be present. As suggested, the sentences pertaining to methodology and results have been interchanged. Kindly verify changes in lines 151, 154-157, 166-167, 173, and 204 to substantiate the modifications.
- The image in Figures B, C, D needs to be enlarged. You should also label the most important elements in your photos so that readers understand the context.
Response to reviewer: Thank you for this valuable suggestion, we have labelled the most important element (presence of microplastic) with white arrows in the figure and explained its meaning in the figure ligand and also enlarged the figure, please verify changes in figure 1
- Remove artificial effects on the edges of photos (Figure 2). It's not scientific. Do not use bold fonts for captions of Figures 2-8.
Response to reviewer: Thank you for your suggestion, we totally agree with it. We have made changes in the figure and removed bold fonts. Please verify the correction in figure 2.
- The authors studied many dozens of preparations of the Drosophila digestive system. The authors saw general patterns of changes in the length and diameter of different parts of the intestine, accessory glands, and excretory system. The reader will not be able to see all this only based on the analysis of Figure 4. I recommend that the authors, in the title of Figure 4, briefly describe how exactly this version of the experiment differs from the control, indicating with numbers the corresponding changes in the photograph. This drawing is worthy of increasing the image size by 100% in both length and width.
Response to reviewer: Thank you for this excellent suggestion and we also appreciate the reviewer 2 for keen observations in the manuscript. In figure of trypan blue staining for gut, we have explained the results in result section before the images as well as in ligand, for more clarity we have labelled the organs of reproductive part in figure 4 which is now making it understandable that which part got affected and stained. Please verify the changes in Line 336- 339, 342- 346
- In the name of both the abscissa and ordinate axis (Fig. 3), you need to put a comma before the unit of measurement, rather than write (g/l) in parentheses. The same remark applies to the labels on the x-axis of Figure 3A. Don't use bold fonts. There is no need to repeat the legend 4 times.
Response to reviewer: Thank you for your valuable suggestion, changes have been made as per comments and we have also removed the bold fonts in all images, please verify changes in image 3, 5 and 6.
- The same remark on Figure 5 and subsequent ones.
Response to reviewer: Thank you for your comment, we have incorporated the Comment in figure 5 also.
- All % in the article (for example, lines 319-321; in addition, three technical errors were also found on these drains) must be rounded to the nearest tenth.
Response to reviewer: Thank you for your suggestion, changes have been incorporated as per comments, please verify changes in line 371-372.
- Genes are written in italics (for example, lines 313, 318, 322 and others).
Response to reviewer: Thank you for your valuable suggestion, we have changed the genes in italics in whole manuscript, please verify the changes in line 25, 26, 249, 250, 253, in table 1, 370, 374, 376, In figure 6A.
- Statistical processing of samples must be carried out again. For example, in Figure 6B. The reader wants to see all the differences between all samples at the 0.05 level. A Tukey test needs to be done here. The authors mention it on line 225, but the results are not reflected in Figures 3, 5, 6.
Response to reviewer: Thank you for your suggestion and we totally agree with your opinion that significance within the all groups must be mentioned. So, we have Incorporated the significance in figures 3, 5, 6 between all samples. Please verify the changes in figures.
- Section The discussion should be divided into subsections. This will make the presentation clearer.
Response to reviewer: Thank you for your suggestion, it will make the discussion part more understandable. We have considered your suggestion in the manuscript please verify the changes in manuscript.
- Figure 8 contains unconfirmed patterns. Be careful when reviewing the literature and displaying the results in a diagram. There is definitely no need to list the diseases of mammals in the picture depicting a fly larva.
Response to reviewer: Thank you for your precise observation and we totally agree with your suggestion. Figure 8 has been changed according to the comment.
- In literature, semicolons should be between authors.
Response to reviewer: Thank you for the suggestion, changes incorporated in the references, please verify changes.
- The periods after the abbreviated names of the magazines are placed carelessly. Not all journal names are abbreviated according to the rules.
Response to reviewer: We apologise for this act of carelessness and thank you for the suggestion, we have incorporated the suggestion in the references, please verify changes
- Spaces between initials are placed carelessly.
Response to reviewer: Thank you for your keen observation, we have incorporated the suggestion in references, please verify the changes.
- Many articles do not have DOIs.
Response to reviewer: Thank you for your suggestion. We have mentioned DOI of the articles in reference section, but some of the publications don’t have DOIs so, we kindly want to inform you that we were unable to mention them.
- Journal titles and volumes are not in italics
Response to reviewer: Thank you for your suggestion, changes have been made in references as per your suggestion as per the comment.

Reviewer 3 Report
Comments and Suggestions for Authors
The aim of this study is to investigate the impact of PET microplastic on Drosophila melanogaster digestive tract and reproductive capacity.
General comment: the general aim of this study is not clearly stated and this paper should be re-written to enhance its quality.
Specific comments:
- The introduction is not appropriate. The objectives of the work should be clearly stated along with a testable hypothesis.
- The abstract is not appropriate. It should state briefly the purpose of the research, the principal results and major conclusions.
- Page 3 line 101, in the M&M section, state the n value for each experiment (how many flyes, how many experiments for each treatment, etc)
- Page 5 line 223, the statistical analysis section should be enhanced and clarified. What kind of software was used, what criteria have you settled and what kind of datum have you used for each anova analysis?
- Page 6, lines 228-232 too long and giving no results, rephrase: “Green fluorescence of PET in the digestive tract of the 231 larvae was confirmed by confocal microscopy (Figure 1)
- Page 6, lines 232-237 this is M&M text
- Page 6, line 239 what kind of microscopy was used to obtain images A1 and A2?
- Page 6, lines 243-249 and throughout the Result sections. No need for extra info here. The results section should only report the obtained results in a clear and concise way, move all other info to M&M.
- Page 7, lines 251-256. Rephrase. Simply use “72 h third instar larvae” to refer to this stage.
- Page 7 line 258, page 8 line 283, page 9 line 297, page 9 line 311, you should start these paragraphs reporting the significance level obtained with you stats (PET MP treatmemt had a significant effect on third instar larvae (Fx = y; p < z)…)
- Page 7 line 288, what % of reduction? Was is significant? Where is the stat?
- Page 7 line 290, this is M&M text
- Page 7 lines 292-295, bad English, please rephrase
- Figure 5, these images are not clear. Be consistent with legends and axis titles.
- Page 9, lines 305-309. Bad English, please rephrase
- Page 9 line 309, be consistent with naming. If group I is control, always name it “control”.
- Page 9 lines 311-324. This paragraph is overall not clear and should be re-written. Line 316 should be the first phrase of this paragraph.
- Page 10 line 328. No need to use all these different symbols for significance.
- Page 10 lines 335-339 and throughout the Discussion paragraph: the discussion section should explore the significance of the results of the work, not repeat them. This text is not appropriate and all this paragraph should be rewritten.
- Page 13 lines 447-450. This should be simply stated as a “dose-dependent effect”.
Comments on the Quality of English LanguageAlthough this research could provide interesting results, the paper is poorly written and must be improved to enhance its quality before publication. Unfortunately, I advice the authors to re-write many paragraphs. There is also some confusion between paragraphs (for ex, results contain M&M text, etc).
Author Response
Reviewer 3
The aim of this study is to investigate the impact of PET microplastic on Drosophila melanogaster digestive tract and reproductive capacity.
General comment: the general aim of this study is not clearly stated and this paper should be re-written to enhance its quality.
Response to reviewer: We appreciate your insightful suggestions, which will aid in improving the overall quality of the manuscript. In response, we have rewritten several paragraphs and highlighted in the revised version. Kindly validate the revisions throughout the manuscript. We have defined the aim in introduction as per your suggestion, please verify the changes in line 94-98
Specific comments:
The introduction is not appropriate. The objectives of the work should be clearly stated along with a testable hypothesis.
Response to reviewer: Thank you very much for your great suggestion. This will strengthen the text and improve its quality. For introduction, we replaced the section about microplastic introduction with information on - why research is required for the selected subject and what research has already been conducted previously. Please verify the changes in Line 52-62 and 95- 100
The abstract is not appropriate. It should state briefly the purpose of the research, the principal results and major conclusions.
Response to reviewer: Thank you for your valuable suggestion, we have changed the abstract content as per your comment. Please verify the changes in line 18-26 and 29-32
Page 3 line 101, in the M&M section, state the n value for each experiment (how many flyes, how many experiments for each treatment, etc)
Response to reviewer: Thank you so much for your suggestion, we kindly want to inform you that in line 113 of revised manuscript we have only mentioned about the formation of microplastic, but n value for each experiment is mentioned in the statistical analysis section, result section (Ligands) and also about the number of flies and larvae used are mentioned in the methodology section. Please verify the suggestion in line 136-138, 204, 326, 362
Page 5 line 223, the statistical analysis section should be enhanced and clarified. What kind of software was used, what criteria have you settled and what kind of datum have you used for each anova analysis?
Response to reviewer: Thank you for this excellent suggestion, we appreciate the reviewer for this keen observation. Additional information is added in the section, please verify the changes in line 266- 268
Page 6, lines 228-232 too long and giving no results, rephrase: “Green fluorescence of PET in the digestive tract of the 231 larvae was confirmed by confocal microscopy (Figure 1)
Response to reviewer: Thank you for the suggestion and we apologies for such discrepancy in sentences. Line has been removed from result section and we have mentioned it in m&m section to provide the reference to support green colour fluorescence of PET microplastics line 156
Page 6, lines 232-237 this is M&M text
Response to reviewer: Thank you for your suggestion, we have replaced this sentence in M&M section, please verify changes in 154- 157
Page 6, line 239 what kind of microscopy was used to obtain images A1 and A2?
Response to reviewer: We appreciate the reviewer for observing such aspects of experimentation details in manuscript for making it more precise. We have now mentioned the microscope for both images line 284
Page 6, lines 243-249 and throughout the Result sections. No need for extra info here. The results section should only report the obtained results in a clear and concise way, move all other info to M&M.
Response to reviewer: I appreciate the insightful suggestion that you have provided. We are delighted to notify you that your recommendations are contributing to the enhancement of the manuscript. The line's content is relocated to lines 166-167 and 173 of the materials and methodology section. Please verify the changes in revised manuscript.
Page 7, lines 251-256. Rephrase. Simply use “72 h third instar larvae” to refer to this stage.
Response to reviewer: Thank you for your valuable suggestion, we have incorporated your suggestion. Please verify the changes in line 295
Page 7 line 258, page 8 line 283, page 9 line 297, page 9 line 311, you should start these paragraphs reporting the significance level obtained with you stats (PET MP treatmemt had a significant effect on third instar larvae (Fx = y; p < z)…)
Response to reviewer:
Thank you for your valuable suggestion. We kindly want to inform you that all the stats are mentioned in the ligands of the figures; thus, we have not repeated the significant difference in paragraphs but according to your suggestion we have made some changes for the starting of paragraph. Please verify changes in Line 302, 349, and 368.
Page 7 line 288, what % of reduction? Was is significant? Where is the stat?
Response to reviewer: Thank you for your great observation. We would like to clarify that the point of view for mentioning the reduction is based on the visible observation of both left and right ovary size, but we totally agree with your query that every result need statistical report. So, to make this statement more appropriate we have changed it in hypothetical way from scientific one. Please verify the correction in Line 337- 339
Page 7 line 290, this is M&M text
Response to reviewer: Thank you so much for valuable suggestion. Following statement has been removed to M&M section, please verify the changes in line 204
Page 7 lines 292-295, bad English, please rephrase
Response to reviewer: We sincerely apologize for language error and thankful to you for notifying us such discrepancies. Sentence has been rephrased according to comment, line 342, 346
Figure 5, these images are not clear. Be consistent with legends and axis titles.
Response to reviewer: Thank you for your valuable suggestion, we totally agree with your point. We have rephrased and mentioned the axis terms in brackets with ligands for better understanding and replaced fertility term from figure 5(a) to reproductive performance, please verify the changes in line 360
Page 9, lines 305-309. Bad English, please rephrase
Response to reviewer: We sincerely apologize for language error and thankful to you for notifying us such discrepancies. We rephrased the sentences as per your suggestion, 359- 365
Page 9 line 309, be consistent with naming. If group I is control, always name it “control”.
Response to reviewer: Thank you for the suggestion. We have taken your suggestion into consideration. Please verify changes in line 364
Page 9 lines 311-324. This paragraph is overall not clear and should be re-written. Line 316 should be the first phrase of this paragraph.
Response to reviewer: Thank you for your valuable suggestion. We have rephrased the whole paragraph and used line 16 for the starting of please verify the changes in line 368- 377.
Page 10 line 328. No need to use all these different symbols for significance.
Thank you for your valuable suggestion and excellent observation. We kindly want to clarify that we have used different symbols of significance for different genes to show the comparison (control v/s gene) otherwise it will be difficult to differentiate the groups.
Page 10 lines 335-339 and throughout the Discussion paragraph: the discussion section should explore the significance of the results of the work, not repeat them. This text is not appropriate and all this paragraph should be rewritten.
Response to reviewer: Thank you so much for your valuables suggestions and yes, we agree with the suggestion and we have removed the unnecessary statements which should be the part of introduction and results from the discussion and rephrased the discussion section with meaningful statements and explaining the significance of the results. Please verify the correction in line 386-418, 431- 435, 439- 448,464- 471
Page 13 lines 447-450. This should be simply stated as a “dose-dependent effect”.
Response to reviewer: Thank you for your suggestion. We have incorporated your suggestion in the manuscript, please verify the change in 487
(Specific comments on English
Although this research could provide interesting results, the paper is poorly written and must be improved to enhance its quality before publication. Unfortunately, I advice the authors to re-write many paragraphs. There is also some confusion between paragraphs (for ex, results contain M&M text, etc).
Response to reviewer: We would like to appreciate the reviewer for providing such insightful suggestions. We have noted this suggestion very seriously and made changes in the manuscript to increase its quality and substantially improved the language, please see revised manuscript for the same.

Round 2
Reviewer 2 Report
Comments and Suggestions for Authors
The manuscript has been sufficiently improved and can be recommended for publication if Figure 3 is enlarged.
Author Response
The manuscript has been sufficiently improved and can be recommended for publication if Figure 3 is enlarged.
Thank you for your valuable comment, we have enlarged the figure 3 as per your suggestion, please verify the changes.

Reviewer 3 Report
Comments and Suggestions for Authors
The paper has now improved and is ready for publication.
Author Response
Thank you for consideration.